# Deep Graph Predictions using Dirac-Bianconi Graph Neural Networks

## Abstract

Viewing Graph Neural Networks as network dynamical systems on graphs has proven a fruitful inspiration for designing interesting GNN architectures. This work introduces the Dirac-Bianconi Graph Neural Network (DBGNN) based on Bianconi's topological Dirac equation on graphs. While heat equations based on network Laplacian tend to smooth out differences, Dirac equations typically feature long-range propagation. Indeed, we find that the DBGNN layer does not lead to an equilibration, or smoothing, of nodal features, even after hundreds of steps. A further distinguishing feature of the topological Dirac equation is that it treats edges and nodes on the same footing. Consequently, we expect DBGNN to be useful in contexts where edges encode more than mere logical connectivity, but have physical properties as well. We show competitive performance on a long-range benchmark dataset for molecular properties, using a roughly 10x smaller model, and superior performance for predicting the dynamic stability of power grids. In the case of power grids, DBGNN achieves robust out-of-distribution generalization, showing that structural relations are learned.

## 1 Introduction

Neural network architectures for graph-structured data have been motivated in various ways (Wu et al., 2021). Spectral approaches see the eigenbasis of the graph Laplacian $L$ as playing a similar role as that of Fourier modes in spatial data. Operating on the representation of the data in the eigenspace of the Laplacian is then seen as an analog of operating on the Fourier transform of data on regular grids.

Spatial convolution approaches instead build on the idea of aggregating features by considering spatial neighborhoods. Message passing is one of the most general implementations of this idea. For the edge connecting the nodes $i$ and $j$, we construct a message $e_{ij}$ from the inputs $x_i^{\text{in}}$, $x_j^{\text{in}}$ at its ends. Then, every node aggregates messages on the edges connected to it to produce an output. Denoting general, potentially non-linear, functions with $f,g,h$, we have

$$e_{ij} = g\left(x_i^{\text{in}}, x_j^{\text{in}}\right), \tag{1}$$

$$x_i^{\text{out}} = f\left(x_i^{\text{in}}, \sum_{n \in \mathcal{N}_n} h(e_{ij}, x_j^{\text{in}})\right). \tag{2}$$

This type of structure resembles that of a dynamical system on a network. Taking this analogy seriously, a rich literature using GNNs inspired by dynamical systems has been developed, often with the hope of overcoming shortcomings in earlier architectures. A notable example is the work by Rusch et al. (2022), who introduce neural networks based on inertial Kuramoto oscillator networks (Kuramoto, 1975; Acebrón et al., 2005; Rodrigues et al., 2016). They argue that over-smoothing, the observation that many GNN architectures tend to average out features across the graph when iterated too deeply, is analogous to synchronization in such systems. The fact that, given a natural parameter condition, the synchronous manifold is unstable for oscillator dynamics then suggests that their architecture should not suffer from over-smoothing.

This is far from the only example of GNN architectures inspired by theoretical physics. Notable works draw from geometric curvature (Topping et al., 2021), discrete dynamical systems (Oono & Suzuki, 2019), ordinary differential equations (Poli et al., 2019), as well as partial differential equations and their discretization schemes (Chamberlain et al., 2021a;b; Eliasof et al., 2021).

In this work, we follow a similar spirit with an eye toward applications for complex dynamical systems. One motivating application is the prediction of the stability properties of power grids. Ringsquandl et al. (2021) show that for power grids, GNNs with 13 or more layers are needed to achieve good performance. This is in contrast to many common benchmark datasets, where GNNs with only 2–3 layers perform best, and it suggests that architectures that tend to over-smooth will struggle in this context. Furthermore, power grids are characterized by node and edge features that are of comparable nature and importance. The same physical laws apply to the edges and nodes. This suggests the need to treat edges and nodes on a similar footing, rather than considering the former as merely a (possibly weighted) coupling for the latter. Finally, we know that complex topological aspects of the graph topology play a role in shaping dynamical properties in power grids, in some cases much more so than features on the nodes. A striking example is shown by Nitzbon et al. (2017), who find that certain desynchronization modes of their power grid model only occur in specific topological settings, irrespective of the node features.

The physical system upon which we will draw for inspiration is the topological Dirac equation on networks, recently introduced by Bianconi (2021). The Dirac equation is one of the foundational equations of quantum mechanics. It describes the evolution of the Dirac field, which describes most elementary particles, such as electrons, protons, and quarks. The Dirac equation is based on the Dirac operator, a square root of the Laplacian. This is obtained by mixing spatial derivatives with transformations in 'internal space', i.e., transformations between different components of the Dirac field. Bianconi (2021) builds on earlier work on quantum information processing by Lloyd et al. (2016) introducing a Dirac operator on simplicial complexes, to introduce a Dirac operator for graphs.

There are two properties of the topological Dirac equation that make it of interest to us. The first is that the topological Dirac equation of Bianconi (2021) is based on treating edges and nodes on the same footing. The internal space of the Dirac field is split between edges and nodes, and Bianconi's Dirac operator mixes the two. The second interesting property is that its square is given by a pair of ordinary graph Laplacians. The graph Laplacian is analogous to a second-order differential operator. The evolution of a density on the nodes under the Laplacian induces diffusion, differences tend to equilibrate. This can be seen as a natural origin of over-smoothing suffered by some GNN architectures. The Dirac operator, on the other hand, is first-order, and we can expect that evolution driven by the Dirac operator induces long-range directional propagation instead.

**Our main contribution is** the development of a Dirac-Bianconi Graph Neural Network (DBGNN) based on a modified generalized Dirac-Bianconi equation. We consider a simple Euler discretization of the Dirac-Bianconi equation, add learnable weights in feature space, add a nonlinearity, and let the resulting equation evolve for several time steps, before reading out the prediction from the evolved features. This enables potential long-range propagation of features that contribute to the prediction. We show that this architecture achieves great performance for a challenging power grid task, and validate it with good performance for molecular property prediction.

The paper is structured as follows. We start by introducing Bianconi's Dirac operator for graphs, as well as the topological Dirac equation and our generalization thereof. This provides the basis for a detailed introduction of the DBGNN layer. Thereafter, we show experimental results and compare the performance to benchmark models, closing with the discussion and further outlook.

**Notation:** Graphs $\mathcal{G}$ consist of nodes $\mathcal{N}$ and edges $\mathcal{E}$. Each edge occurs twice, with the two possible orientations, and we write an edge $e$ as an ordered pair $[i, j] \in \mathcal{E}$ of nodes $i, j \in \mathcal{N}$. The set of neighbors of node $i$ is denoted as $\mathcal{N}_i$. The space of features on an edge/node is called $F_{e/n}$, the space of all edge features of our graph $\mathcal{G}$ is $F_e^{\mathcal{E}}$, and $F_n^{\mathcal{N}}$ for node features.

## 2    DIRAC OPERATOR FOR GRAPHS

In differential geometry, any square root of the Laplacian (typically on a vector bundle over a Riemannian manifold) is called a Dirac operator. While the eigenvalues of the Laplacian operator show how diffusive processes disperse, they do not distinguish different directions in the manifold. In contrast, those of the Dirac operators typically do so by coupling directions in space to different directions in the bundle. This can be illustrated with the simplest example. For the tangent bundle over $\mathbb{R}$, the Dirac operator $-i\partial_x$ has eigenvalues $\pm 1$ for the eigenmodes $e^{\pm it}$. The Laplace operator has the same eigenmodes, but both correspond to the eigenvalue 1. Exponentiating the Laplacian leads to the heat kernel, which smooths out differences and leads to equilibration in the long run: All states converge to the kernel of the Laplace operator. In contrast, exponentiating $-i\partial_x$ simply induces shifts along the real axis: $e^{-si\partial_x} f(x) = f(x - s)$, as can be readily verified by taking the Fourier transform.

The structure of a graph can be described using the incidence matrix:

$$B_{ie} = \begin{cases} +1 \text{ if } e = [i, j] \\ -1 \text{ if } e = [j, i] \\ 0 \text{ otherwise.} \end{cases} \tag{3}$$

In the context of homological algebra this is called the boundary operator. It is a straightforward calculation to see that $BB^\dagger$ is the usual Laplacian matrix for the graph. $B^\dagger B$ is the so-called one-down Laplacian and connects edges to edges. These boundary operators play a central role in the extension of GNNs to simplicial complexes, a principled approach to give an analogy of the message passing framework to this setting is given in Bodnar et al. (2021).

In our context, the relationship between incidence matrix and Laplacian suggests that the incidence matrix should play a role similar to that of the Dirac operator. However, as it maps between edges and nodes, it cannot be used directly to update features. Furthermore, it is not immediately obvious how to interpret the expectation that different directions should couple to different internal states. A graph does not by itself provide a notion of direction that extends beyond a single edge.

The insight of Bianconi (2021) is that this can be overcome by considering edges and vertices on an equal footing. As $B$ maps from the edge space to the node space, and $B^\dagger$ vice versa, this allows to introduce a natural Dirac operator:

$$\partial_{DB} = \begin{pmatrix} 0 & bB \\ (bB)^\dagger & 0 \end{pmatrix} \tag{4}$$

where $b \in \mathbb{C}$ is some complex number. This is illustrated in Figure 1a. The square of $\partial_{DB}$ is then block diagonal, with the usual and the one-down Laplacian on the diagonal:

$$\partial_{DB}^2 = 2|b|^2 \begin{pmatrix} BB^\dagger & 0 \\ 0 & B^\dagger B \end{pmatrix} . \tag{5}$$

If we attach a feature $x_i, e_{ij} \in \mathbb{R}$ to each node and each edge respectively, and introduce the notation, $e_{ji} = -e_{ij}$ for all edges $[i, j] \in \mathcal{E}$, the Dirac-Bianconi operator acts simply as

$$\partial_{DB} \begin{pmatrix} x \\ e \end{pmatrix} = \begin{pmatrix} x' \\ e' \end{pmatrix} \qquad \text{with} \qquad \begin{matrix} x_i' = b \sum_{j \in \mathcal{N}_i} e_{ij} \\ e_{ij}' = b^*(x_i - x_j) \end{matrix}. \tag{6}$$

Thus, the edge feature space serves to encode directional information on the state of the features of the original graph. In this form, the operator still resembles the action of a standard message-passing architecture.

As $\partial_{DB}$ squares to the usual graph Laplacians, iterating this operator does not introduce interesting non-diffusive dynamics on the feature space of the graph by itself, however the dynamical equations induced by it are different to those obtained with the Laplacian. The concrete equation considered by Bianconi (2021) includes a mass term $\beta$. Then letting edge and nodes states depend on $t \in \mathbb{R}$, the topological Dirac equation is given by

$$i\partial_t \begin{pmatrix} x \\ e \end{pmatrix} = \left( \partial_{DB} + \begin{pmatrix} \beta & 0 \\ 0 & -\beta \end{pmatrix} \right) \begin{pmatrix} x \\ e \end{pmatrix} \tag{7}$$

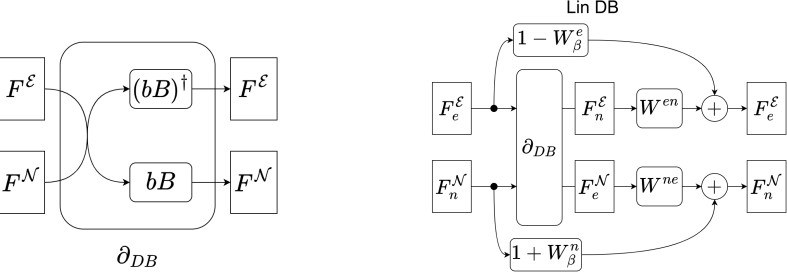

(a) Action of Bianconi's Dirac Operator.     (b) Generalized Dirac Bianconi dynamics.

Figure 1: Linear dynamics mixing node and edge spaces, using the Dirac operator for networks.

This is a wave equation in the sense that the right-hand side is Hermitian, and thus the time evolution has purely imaginary eigenvalues. However, this equation does not capture directionality in the same way as the usual Dirac operators of quantum mechanics. For example, on a regular lattice, the operator does not distinguish edges that are parallel or orthogonal to each other. However, it does mix edge features and node features in non-trivial ways. In addition, its eigenstates encode topological features of the graph on both the edges and the nodes.

Following the spectral analysis of Bianconi (2021), we can also see that this evolution cannot induce equilibration in the same way as Laplacian evolution does. The Hermitian operator $\partial_{DB} + \mathrm{diag}(\beta, -\beta)$ has positive and negative eigenvalues that are bounded away from 0 by $|\beta|^2$, thus it has no kernel and no steady state to which one could converge. As long as $\beta$ is non-zero, dynamics based on $\partial_{DB} + \mathrm{diag}(\beta, -\beta)$ will always have decaying and expanding directions in feature space. When multiplied with the imaginary unit, these are turned into rotating and counterrotating oscillations. This is similar to the considerations in Rusch et al. (2022), which are based on the instability of the homogeneous state (typically known as 'synchronous state' for oscillators).

We consider the Euler discretization of equation 7 and introduce higher-dimensional feature spaces of dimension $d_n$ for nodes and dimension $d_e$ for edges, $F_n = \mathbb{R}^{d_n}$ and $F_e = \mathbb{R}^{d_e}$. Consider coupling matrices $\boldsymbol{W}^{ne} \in F_n \otimes F_e$ and $\boldsymbol{W}^{en} \in F_e \otimes F_n$, and mass matrices $\boldsymbol{W}_\beta^n \in F_n \otimes F_n$, $\boldsymbol{W}_\beta^e \in F_e \otimes F_e$, we can write

$$x_i(t+1) = x_i(t) + \boldsymbol{W}^{ne} \sum_{j \in \mathcal{N}_i} e_{ij}(t) + \boldsymbol{W}_\beta^n x_i(t), \tag{8}$$

$$e_{ij}(t+1) = e_{ij}(t) + \boldsymbol{W}^{en}(x_i(t) - x_j(t)) - \boldsymbol{W}_\beta^e e_{ij}(t).$$

We call this equation the generalized linear Dirac-Bianconi equation, illustrated in Figure 1b. Here $F_n^{\mathcal{N}} = \mathbb{R}^{d_n|\mathcal{N}|}$ denotes the total node feature space, and $F_e^{\mathcal{E}}$ the total edge feature space. However, after applying $\partial_{DB}$ we also have the space of one edge feature space per node, $F_e^{\mathcal{N}}$ and one edge feature space per edge $F_n^{\mathcal{E}}$. The action of $\boldsymbol{W}^{ne}$ and $\boldsymbol{W}^{en}$ then maps us back to the original $F_n^{\mathcal{N}}$ and $F_e^{\mathcal{E}}$.

This equation contains both the wave behavior, as well as expanding/contracting dynamics as a special case. The oscillatory behavior occurs if we mimic the imaginary unit by making the right-hand side of equation 8 antisymmetric: $\beta_{n/e} = -\beta_{n/e}^\dagger$, and $\boldsymbol{W}^{ne} = -\boldsymbol{W}^{en\dagger}$.

This should be compared to a simple linear MPNN style dynamics with edge weights:

$$x_i(t+1) = x_i(t) + \boldsymbol{W}_n^{\mathrm{MPNN}} \sum_{j \in \mathcal{N}_i} e_{ij}(t) + \beta_n x_i(t), \tag{9}$$

$$e_{ij}(t) = \boldsymbol{W}_e^{\mathrm{MPNN}}(x_i(t) - x_j(t))$$

while the edge messages change in time, there is no time evolution of edge features themselves. The messages can trivially be eliminated, while this is not the case for equation 8. This changes the way the dynamics spreads into the network, see Figure 2. Notably, this is true, even if the dataset has no non-trivial edge features.

The equation 8 is of the general form considered in Bodnar et al. (2021), however, the specific form here has not been considered there. From the perspective of simplicial topology, the key

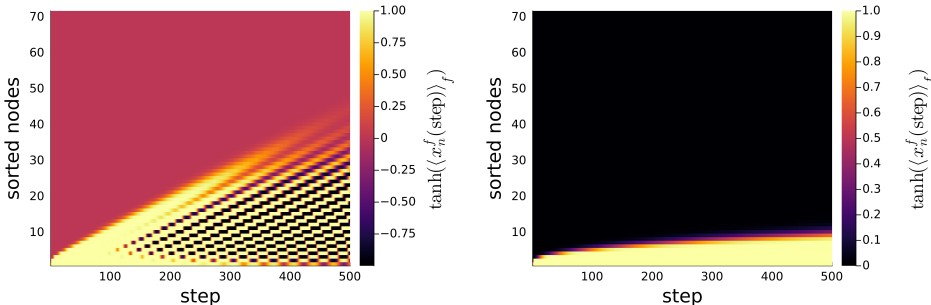

Figure 2: Feature activation versus steps of the linear DB equation 8 (left) and the MPNN equation 9 (right) on a path graph, $d_n = d_f = 1$, same random weights. The initial condition has all features zero, except at node 1 where the node features are randomly activated. Linear DB shows activation traveling down the graph linearly while MPNN shows diffusion.

differentiator is that we use the boundary and coboundary operator, together with a simultaneous update of nodes and edges, but no Laplacian. For simplicial two-complexes this has been considered in Bunch et al. (2020), but we are not aware of prior work that takes this approach for graphs.

## 3  DIRAC-BIANCONI GRAPH NEURAL NETWORKS

In the following, we will use the generalized linear Dirac-Bianconi equation to define a novel GNN layer for problems in which long-range interactions in the graph are expected to play a profound role, and where node features and edge features are on a similar footing.

Based on the above considerations, we define the DB 1-Step layer as one step of the linear DB equation 8, followed by a dropout and a nonlinearity. The matrices $\boldsymbol{W}^{ne} \in F_n \otimes F_e$, $\boldsymbol{W}^{en} \in F_e \otimes F_n$, $\boldsymbol{W}_\beta^n \in F_n \otimes F_n$ and $\boldsymbol{W}_\beta^e \in F_e \otimes F_e$ are learnable weights. This layer is sketched in Figure 3 A). We concatenate $T$ such layers with shared weights to obtain the DB T-Step layer of Figure 3 B). The full Dirac Bianconi Graph Neural Network (DBGNN) with $K$ T-Step layers then is constructed as in Figure 4:

- First we map the input features linearly to the hidden feature spaces $F_n = \mathbb{R}^{d_n^{\text{hidden}}}$ and $F_e = \mathbb{R}^{d_e^{\text{hidden}}}$ for nodes and edges respectively.

- Then we alternate DB T-step layers and skip connections that mix in the input features using a linear map $K$ times. This allows different dynamics that see both, the initial conditions as well as the features processed by the previous layers.

- Finally, use MLP maps from the hidden features dimension to the output dimension, optionally followed by pooling and another MLP layer.

Such a DBGNN makes $KT/2$ node to node hops on the graph.

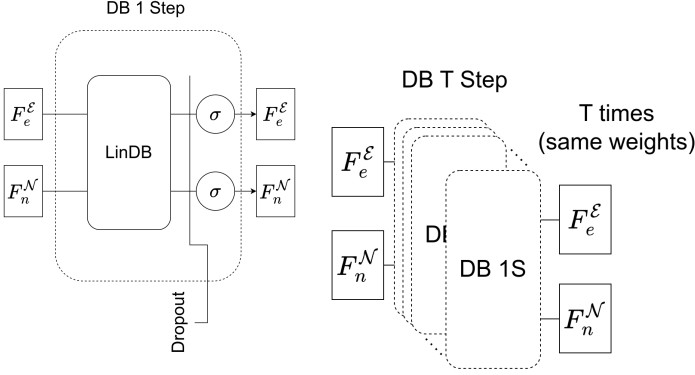

Figure 3: A) Dirac Bianconi 1-Step, B) Dirac Bianconi T-Step

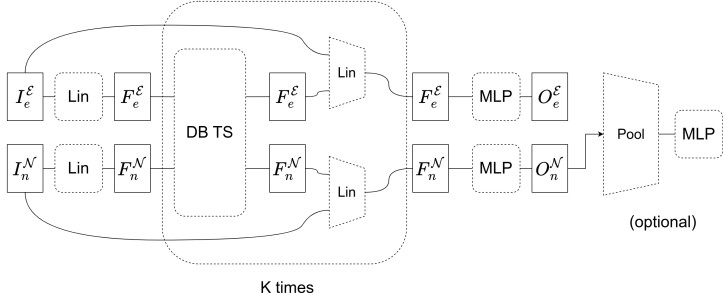

Figure 4: Dirac Bianconi Graph Neural Network (DBGNN)

**Evolution of Dirichlet energy of DBGNN**  A common way to understand whether an architecture is suffering from over-smoothing, is to study the various variations of the Dirichlet energy (Zhou et al., 2021; Wang et al., 2022; Rusch et al., 2022; Chen et al., 2023; Liu et al., 2023; Fu et al., 2023; Di Giovanni et al., 2023), its definition and computation is given in Appendix A.2. To understand the intrinsic equilibration properties of the DBGNN when compared to convolutional GNNs we evaluated the DE for roughly 500 steps of untrained networks. The result is shown in Figure 5. From the spectral analysis of equation 7, we find that no equilibration occurs for DBGNNs, while GCNs quickly lose heterogeneity.

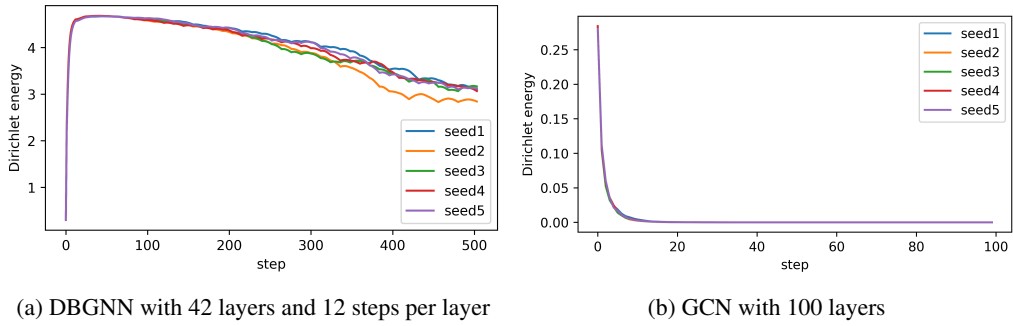

(a) DBGNN with 42 layers and 12 steps per layer      (b) GCN with 100 layers

Figure 5: Evolution of normalized Dirichlet energy of node feature embeddings for a sample of dataset20 with five different seeds and without any training.

**Long-range capabilities, the role of edge asymmetry and non-linearities**  To analyze the long-range capabilities of DBGNN, we analyze how the layers can spread a localized feature into a graph. There are two aspects in comparison to the MPNNs that could enable deep spreading, one is the intrinsic wave dynamics of the linear DB equation, the other is that we apply a non-linearity to the edges. This second aspect resembles the approach of Bodnar et al. (2022). It is notable that with a ReLU activation function on the edges, either $x_i - x_j$ or $x_j - x_i$ is completely suppressed. This induces a strong directionality into the behavior of the layer which could enable long-range propagation.

To investigate the effect of the edge non-linearity and the edge updating separately, we will compare equation 8 and the iterated DB 1-Step layer, with equation 9 and an MPNN equation 10 with and without edge non-linearity:

$$x_i(t+1) = \sigma(\boldsymbol{W}^{ne} \sum_{j \in \mathcal{N}(i)} e_{ij} + \boldsymbol{W}_\beta^n x_i(t)) \qquad (10)$$

$$e_{ij}(t) = \sigma(\boldsymbol{W}^{en}(x_i(t) - x_j(t))) \quad \text{or} \quad e_{ij}(t) = \boldsymbol{W}^{en}(x_i(t) - x_j(t)),$$

for the same randomly initialized weights drawn from a normal distribution with spread 0.1. As discussed in the introduction, equation 8 can induce both oscillatory and non-oscillatory behavior depending on the weights. We here investigate the properties of the oscillatory regime, in which we might expect propagating waves. To do so we constrain the weights to be $\boldsymbol{W}^{ne} = -\boldsymbol{W}^{en\dagger}$ and $\boldsymbol{W}_\beta^{n/e}$ antisymmetric.

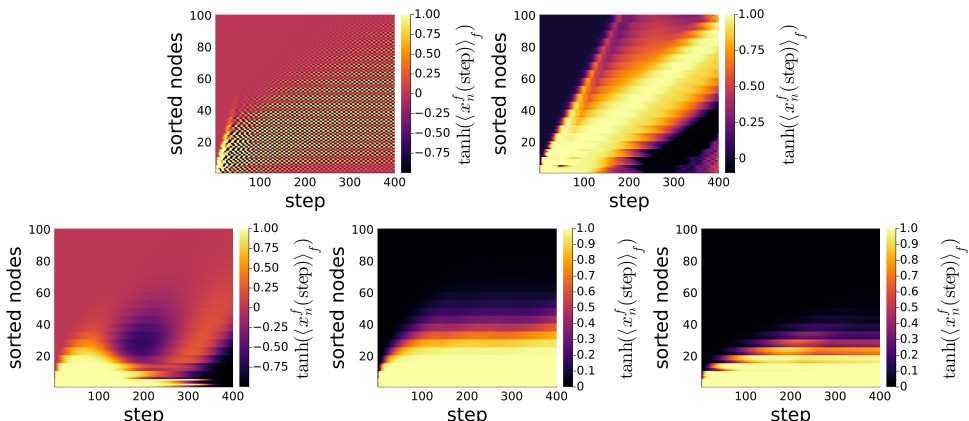

Figure 6: Oscillatory regime: Feature activation versus steps of the linear DB equation 8 (top left), the non-linear DB 1-Step layer equation 8 + ReLU (top right), the linear MPNN layer equation 9 (bottom left), and MPNN equation 10 without (middle) and with non-linear messages (bottom right). $d_n = d_f = 4$, same random weights.

We evolve these models on a 5x20 graph, where one of the short edges has all nodes initialized randomly, and all other edge and node features are identically zero. Figure. 6 shows exemplary trajectories for a model with four edge and node features. The linear DB equation shows a leading edge, a concentrated wave of activation that spreads quickly into the network before dissipating, with ripples radiating into the rest of the network. Due to the oscillatory initialization, the linear MPNN equation also shows oscillatory behavior; however, this does not lead to spreading into the network. Adding the non-linearities stabilizes the leading edge of the DB equation which now reaches the other end of the graph, and also sharpens the ripples into a coherent excitation that travels slower down the graph. For MPNN the nonlinearities suppress the oscillations, we are left with pure diffusion. For higher dimensional internal spaces, as well as for many non-oscillatory random weights, most configurations of all layers exhibit slow diffusion into the system. Occasionally, we can randomly generate coherent travelling excitations in DBGNN, they are not observed in MPNN. We provide examples of these trajectories in the appendix. We conclude from this that the wave aspects of DBGNN enable deep propagation of signals into the graph, while the edge non-linearities play a minor role.

## 4 EXPERIMENTAL RESULTS

We test our DBGNN on three hard tasks related to complex systems: one on power grids, two on molecular structure. The power grid dataset is especially challenging because the topological properties across large distances are of great importance, whereas there are no edge features and only one nodal features. On the contrary, the molecular datasets consist of a variety of nodal and edge features.

**Dynamic stability of power grids**   The most sophisticated dataset dealing with the dynamic stability of power grids is published in Nauck et al. (2023), which is based on Nauck et al. (2022b;a). There is a total of 20,000 grids: 10 000 small grids of size 20 (dataset20), and 10 000 medium-sized grids of size 100 (dataset100). Besides training and evaluating the models on the same grid sizes, we also analyze the out-of-distribution generalization by training the models on grids of size 20 and evaluating them on grids of size 100. We refer to this task as tr20ev100. Training models on smaller grids and evaluating them on larger grids is important for real-world applications, because the computational effort increases at least quadratically with the size of the grid.

The results are provided in Table 1, where we compare them to the current benchmark performances. DBGNN achieves the best performance at all tasks and significantly outperforms the other models at the out-of-distribution generalization. One of the reasons for the superior performance might be related to the capability of going deep without encountering the problem of over-smoothing. The final DBGNN consists of 4 DB 12-step layers, resulting in 48 total steps. To investigate the absence of smoothing further, we compute the Dirichlet energy for one sample of dataset20 in the forward pass using the node embeddings at each step. Figure 7 shows that in trained models, the Dirichlet energy stays high throughout the forward pass, confirming the intuition that DBGNNs do not suffer

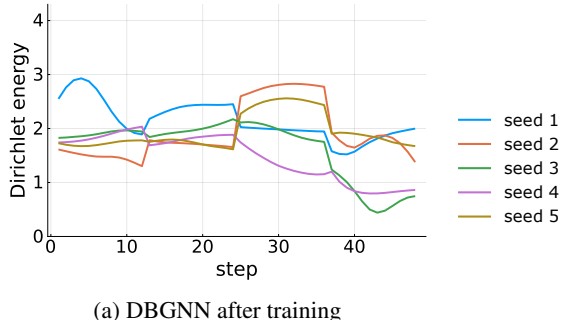

(a) DBGNN after training

Figure 7: Evolution of normalized Dirichlet energy of node feature embeddings in a trained DBGNN layer for a sample of dataset20 with five different seeds.

from oversmoothing even at considerable depth. Some seeds go through periods of considerable "sharpening", especially following the change of dynamics after every T steps.

Table 1: Performance of SNBS prediction measured by $R^2$ score in % using the benchmark models from Nauck et al. (2023).The column *tr20ev20* and *tr100ev100* denotes that the models are trained and evaluated on the same datasets. Out-of-distribution performance is measured by evaluating the models on dataset100 after training them on dataset20 (tr20ev100).

| Model | In-distribution tasks | | Out-of-distribution task |
|---|---|---|---|
| | tr20ev20 | tr100ev100 | tr20ev100 |
| ArmaNet | $82.22 \pm 0.12$ | $88.35 \pm 0.12$ | $67.12 \pm 0.80$ |
| GCNNet | $70.74 \pm 0.15$ | $75.19 \pm 0.14$ | $58.24 \pm 0.47$ |
| TAGNet | $82.50 \pm 0.36$ | $88.32 \pm 0.10$ | $66.32 \pm 0.74$ |
| DBGNN | $\mathbf{84.16} \pm 0.25$ | $\mathbf{88.99} \pm 0.24$ | $\mathbf{72.75} \pm 0.63$ |

**Binding affinity prediction** Predicting protein-ligand binding affinity is one of the challenging and time-consuming tasks in the early stages of drug discovery. Accurate and robust binding affinity prediction algorithms are needed to speed up the process of identifying potential drug candidates by screening extensive ligand libraries for a given protein target. At present, several deep learning-based models are proposed to predict the protein–ligand binding affinity and achieve good performance. While deep learning models, especially graph neural networks methods (Wang et al., 2023; Gorantla et al., 2023; Jiang et al., 2020) have shown promise, however, they suffer from generalizability issues (Gorantla et al., 2023).

Here, we evaluate our DBGNN on the binding affinity prediction task using a graph-based deep learning framework similar to earlier works (Gorantla et al., 2023; Jiang et al., 2020) on a publicly available dataset known as Davis (Davis et al., 2011). Previously, (Jiang et al., 2020) studied both Graph convolutional network (GCN) and graph attention network (GAT) role in binding affinity prediction task and showed that GCN performed better than GAT on the Davis (Davis et al., 2011) dataset. We use a refined Davis dataset used by Gorantla et al. (2023) comprising 22 644 binding interactions of 333 protein targets and 68 ligands. The graph-based deep learning framework for binding affinity prediction takes protein sequence and ligand Simplified Molecular Input Line Entry System (SMILES) string as input and then converts these input sequences and SMILES data into protein and ligand graphs, respectively. The protein and ligand graphs are passed through GNNs to extract features and obtain encodings. These encodings are then combined and passed through a fully-connected neural network for binding affinity prediction. For ligands, the graphs are obtained from SMILES string which is a linearised version of the chemical structure, with atoms as nodes and bonds as edges. In the case of protein sequences, graphs are constructed from contact maps which contain information on which amino acids in the protein sequence are in contact or not. The Pconsc4 (Michel et al., 2019) contact map prediction algorithm is used to obtain protein graphs, with amino acid residues in the sequence as nodes, and their contact information is present in the edges. We keep the framework proposed by Gorantla et al. (2023); Jiang et al. (2020) and only replace 3 GCN layers with 1 DBGNN layer for extracting protein and ligand features. We perform a short hyperparameter study optimizing the learning rates and show the properties of the final configuration in Table 5.

Table 2: Performance comparison of DBGNN and GCN models on binding affinity prediction task.

| Model | Spearman | CI | Pearson | RMSE |
|---|---|---|---|---|
| DBGNN (1 layer) | $0.67 \pm 0.02$ | $0.88 \pm 0.01$ | $0.83 \pm 0.02$ | $0.47 \pm 0.02$ |
| GCN (3 layers) | $0.68 \pm 0.02$ | $0.88 \pm 0.01$ | $0.83 \pm 0.02$ | $0.47 \pm 0.03$ |

**Peptide property prediction**  To assess the performance of DBGNN model on long-range interactions, we use *Peptides-struct* dataset from the long-range benchmark dataset Dwivedi et al. (2022b). Peptides, defined as short chains of amino acids and play crucial roles in numerous biological processes. Given the intricate relationships between peptides and their biological functions, computational prediction of peptide properties is crucial for advancing drug development, and biomolecular engineering. The *Peptides-struct* dataset is a multi-label graph regression task using the 3D structure of peptides. The primary objective here is to predict aggregated 3D properties of peptides at the graph level. These properties include inertia mass, inertia valence, length, sphericity, and plane best fit. These properties have been normalized to a zero mean and unit standard deviation for consistency. *Peptides-struct* encompasses 15,535 graphs, with an average of 150.94 nodes per graph and an average diameter of 56.99. We compare the performance to GINE and GCN-based models from Dwivedi et al. (2022b) in Table 3. The results of better performing transformer-based models are given in the appendix Table 7. The used DBGNN has 63 911 parameters, whereas the GCN models have about 500 000. Nevertheless, DBGNN outperforms the other models.

Table 3: Performance comparison on long-range benchmark dataset *Peptides-struct*. The performances are taken from Dwivedi et al. (2022a)

| model | train MAE | test MAE |
|---|---|---|
| GCN | $0.2939 \pm 0.0055$ | $0.3496 \pm 0.0013$ |
| GCNII | $0.2957 \pm 0.0025$ | $0.3471 \pm 0.0010$ |
| GINE | $0.3116 \pm 0.0047$ | $0.3547 \pm 0.0045$ |
| GatedGCN | $0.2761 \pm 0.0032$ | $0.3420 \pm 0.0013$ |
| GatedGCN+RWSE | $0.2578 \pm 0.0116$ | $0.3357 \pm 0.0006$ |
| DBGNN | $0.2868 \pm 0.013$ | $0.3288 \pm 0.0046$ |

## 5 Conclusion

We introduced a new graph neural network layer based on a straightforward generalization of the topological Dirac-Bianconi equation on networks. We show that this model has no intrinsic tendency to equilibrate features on the network. This has the potential to enable handling long-range dependencies, and to treat edge and node features on an equal footing. By incorporating multiple steps with weight sharing within one layer we enable the layer to efficiently learn dynamics that probe the graph deeply. The DBGNN is a straightforward adaptation of the topological Dirac-Bianconi equation, and thus offers the potential for many further modifications. In its current shape, DBGNN already outperforms other layers at predicting the dynamic stability of power grids and considerably improves performance for out-of-distribution generalization. Further, DBGNN also achieves competitive performance for molecular property predictions.

By analyzing the internal node embeddings using the Dirichlet energy, we can show that DBGNN appears to be not suffering from the over-smoothing problem. Further, when the dynamics change after taking many steps, we observe a sudden sharpening of features, a phenomenon that remains to be better understood. We have shown evidence that the long-range capabilities result from the underlying Dirac Bianconi dynamics, rather than the edge non-linearity.

The experiments as performed were conducted without adapting the model to the task at hand. As DBGNN is close to standard MPNN style networks, the vast array of modifications that exist for these can be adapted in a straightforward manner here. It remains to be seen if they can also enhance DBGNN style networks. One particularly interesting question is, whether long-range modifications to GCNs, such as Gutteridge et al. (2023), can further enhance the long-range behavior of DBGNNs.

Overall the expanded long-range capabilities present newfound opportunities in the domains of power grid analysis and molecular predictions, offering a heightened potential for scientific exploration and understanding.

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

# A  APPENDIX

## A.1  DATA AVAILABILITY

All data and code to train the models and generate the figures will pre provided upon publication on GitHub and Zenodo. The supplementary material to this submission contains code and data to train DBGNN on the power grid dataset of 20 nodes.

## A.2  DIRICHLET ENERGY

The Dirichlet energy is a measure of the heterogeneity of features across the graph. The normalized Dirichlet energy (DE) is computed by:

$$\text{Dirichlet energy} = \frac{\text{tr}(x(k)^\top \boldsymbol{L} x(k))}{\text{tr}(x(k)^\top x(k))}, \tag{11}$$

where $x(k)$ denotes the node embedding after $k$ steps, and tr denotes the trace operator. The interpretation becomes apparent by rewriting the Dirichlet energy in terms of the edge differences:

$$\text{Dirichlet energy} = \frac{\sum_{(i,j)\in\mathcal{E}} ||x^i(k) - x^j(k)||^2}{\sum_{i\in\mathcal{N}} ||x^i(k)||^2}, \tag{12}$$

where $x^i(k) \in F_n$ denotes the state of node $i$ after $k$ steps.

## A.3  HYPERPARAMETERS FOR REPRODUCIBILITY

The following tables provide the hyperparameters to reproduce the main results. For the power grid dataset, the information is given in Table 4, for the binding affinity task, the information is given in Table 5, and for the peptides structure task in Table 6.

Table 4: Properties of DBGNN after hyperparameter studies for power grid datasets

| parameter | dataset with grids of size 20 | dataset with grids of size 100 |
|---|---|---|
| number of layers (K) | 4 | 4 |
| number of steps (with shared weights) per layer (T) | 12 | 12 |
| $d_n^{\text{hidden}}$ | 500 | 500 |
| $d_e^{\text{hidden}}$ | 10 | 10 |
| dropout for node convolution | $6.1 \times 10^{-2}$ | $4 \times 10^{-2}$ |
| dropout for edge convolution | 0 | 0 |
| batch size | 250 | 250 |
| epochs | 10 000 | 5 000 |
| learning rate (LR) | $\approx 3.594 \times 10^{-4}$ | $6 \times 10^{-4}$ |
| scheduler: LR decay factor | 0.65 (period: 500 epochs) | 0.7 (period: 500 epochs) |

## A.4  IMPLEMENTATION AND COMPUTATION DETAILS

For the Julia implementation of DBGNN (Bezanson et al., 2017) the packages *GraphNeuralNetworks.jl*(Lucibello, 2023) and *Flux.jl* (Innes et al., 2018) are used. Furthermore, *Cuda.jl* (Besard et al., 2019) and *MLDatasets.jl* are used. Furthermore, we provide a PyTorch Paszke et al. (2019) implementation using PyTorch Geometric Fey & Lenssen (2019). We run all experiments on NVIDIA V100 accelerators.

## A.5  POWER GIRD EXPERIMENTAL DETAILS

The dynamics of power grids feature complex collective phenomena extending across the whole system (Witthaut et al., 2022). The chosen task is based on the so-called single-node basin stability

Table 5: Properties of DBGNN after hyperparameter studies for binding affinity task

| parameter | value |
|---|---|
| number of layers (K) | 1 |
| number of steps per layer (T) | 10 |
| $d_n^{\text{hidden}}$ (ligand graph) | 156 |
| $d_n^{\text{hidden}}$ (protein graph) | 108 |
| $d_e^{\text{hidden}}$ | 2 |
| dropout for node convolution | 0.2 |
| dropout for edge convolution | 0 |
| batch size | 128 |
| epochs | 6000 |
| learning rate (LR) | 0.0001 |

Table 6: Properties of DBGNN after hyperparameter studies for peptides structure datasets

| parameter | value |
|---|---|
| number of layers (K) | 1 |
| number of steps (with shared weights) per layer (T) | 24 |
| $d_n^{\text{hidden}}$ | 100 |
| $d_e^{\text{hidden}}$ | 100 |
| dropout for node convolution | $1 \times 10^{-2}$ |
| dropout for edge convolution | $1 \times 10^{-2}$ |
| batch size | 20 |
| epochs | 3 000 |
| learning rate (LR) | $\approx 6.608 \times 10^{-5}$ |
| scheduler: LR decay factor | None |

originally introduced by Menck et al. (2013) which describes the nodal dynamic stability. It is the result of expensive Monte-Carlo simulations and quantifies the probabilistic behavior of the entire power grid after applying nodal perturbations.

For the power grid models, the nodal input features are categorical representations of sources or sinks. The power lines are considered homogeneous; hence, the input edge features are simply set to 1. The absence of diverse features puts the focus on topological properties.

The datasets both contain individual train, validation and test sets (70:15:15). The only input feature per node, describing if a node is considered to be a source ($P = 1$) or sink ($P = -1$) is based on the injected power $P$. Since homogeneous coupling is used, there are no edge features. The performance on the nodal regression setup is evaluated using the coefficient of determination ($R^2$).

**Training details**  Different hyperparameters are investigated to identify promising configurations. Among others, we explore the depths and widths of the models and also vary batch size, learning rate, scheduling, and dropout. DBGNNs contain $K$ different DB T-step layers. The steps within each of the DB T-step layers share weights. In the case of the power grids, we did not find skip connections helpful. The resulting configurations of the hyperparameter studies are provided in Table 4.

## A.6 BINDING AFFINITY EXPERIMENTAL DETAILS

We use a refined Davis dataset used by Gorantla et al. (2023) comprising 22 644 binding interactions of 333 protein targets and 68 ligands. The graph-based deep learning framework for binding affinity prediction takes protein sequence and ligand Simplified Molecular Input Line Entry System (SMILES) string as input and then converts these input sequences and SMILES data into protein and ligand graphs, respectively. The protein and ligand graphs are passed through GNNs to extract features and obtain encodings. These encodings are then combined and passed through a fully-connected neural network for binding affinity prediction. For ligands, the graphs are obtained

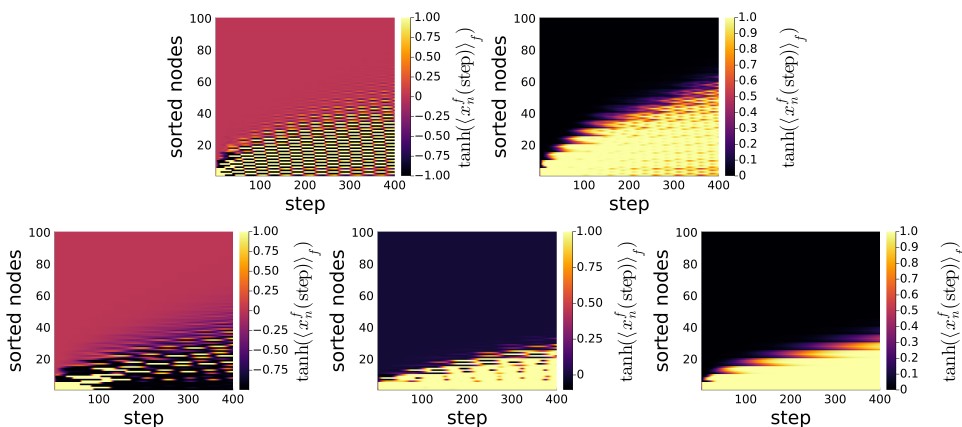

Figure 8: Non-oscillatory regime: Feature activation versus steps of the linear DB equation 8 (top left), the non-linear DB 1-Step layer equation 8 + ReLU (top right), the linear MPNN layer equation 9 (bottom left), and MPNN equation 10 without (middle) and with non-linear messages (bottom right). $d_n = d_f = 4$, same random weights.

from SMILES string which is a linearized version of the chemical structure, with atoms as nodes and bonds as edges. In the case of protein sequences, graphs are constructed from contact maps which contain information on which amino acids in the protein sequence are in contact or not. The Pconsc4 (Michel et al., 2019) contact map prediction algorithm is used to obtain protein graphs, with amino acid residues in the sequence as nodes, and their contact information is present in the edges. We keep the framework proposed by Gorantla et al. (2023); Jiang et al. (2020) and only replace 3 GCN layers with 1 DBGNN layer for extracting protein and ligand features. We perform a short hyperparameter study optimizing the learning rates and show the properties of the final configuration in Table 5.

## A.7 PEPTIDES STRUCTURE EXPERIMENTAL DETAILS

The second molecular dataset contains node as well as edge features and long-range dependencies. Given the nature of the task, namely predicting peptide properties, longer range interactions are expected to be relevant. Table 7 contains the results of transformer-based GNNs for the peptides task.

Table 7: Performance comparison on longe-range benchmark dataset *Peptides-struct*. The results are from Dwivedi et al. (2022a).

| model | train MAE | test MAE |
|---|---|---|
| Transformer+LapPE | $0.2403 \pm 0.0066$ | $\mathbf{0.2529} \pm 0.0016$ |
| SAN+LapPE | $0.2822 \pm 0.0108$ | $0.2683 \pm 0.0043$ |
| SAN+RWSE | $0.2680 \pm 0.0038$ | $0.2545 \pm 0.0012$ |

## A.8 NON-OSCILLATORY RANDOM WEIGHTS

Figure 8 provides exemplary trajectories with random inital weights for which MPNN and DBGNN do not differ substantially. We suspect that random initialization leads to a washing out of the directionality required to generate coherent propagation. How to generate travelling activation robustly remains an open research question.

We also provide a supplementary movie that shows how a DBGNN layer can have activation travelling along a ladder graph, and being reflected at the far edge. The trajectory is provided in Figure 9.

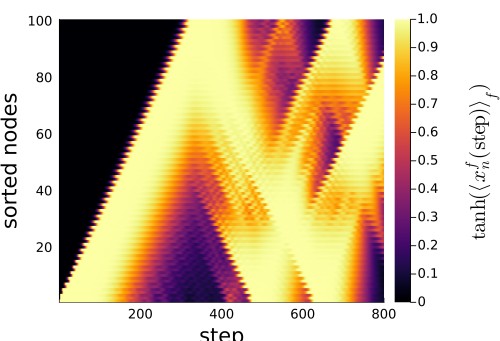

Figure 9: Oscillatory regime, full DB 1-Step layer equation 8 + non-linearity, ladder graph, $d_n = d_f = 4$.

