# OpenReview forum: "Deep Graph Predictions using Dirac-Bianconi Graph Neural Networks"
_ICLR.cc/2024/Conference — Submitted to ICLR 2024_

### Official Review · Reviewer_3z7c · 2023-10-26

**Soundness:** 4 excellent
**Presentation:** 4 excellent
**Contribution:** 3 good
**Rating:** 6
**Confidence:** 4

**Summary:**

Edit: I have increased my score in light of your responses.

It is has been noted that the message passing operations in standard GNNS are essentially equivalent to heat propagation over the network. This can lead to desirable smoothing on relatively shallow networks. However, letting heat propagation proceed for too long leads to oversmoothing and loss of information. Therefore, the authors instead proposed an alternative approach based on the Dirac-Biaconi equation.

The authors cite as an example GNNs aiming to predict the stability of power grids. In this setting 13 layers are needed. It is also to treat nodes and edges as equally important.

The author utilized the Dirac-Bianconi operator which was previously introduced (in the graph setting) by Bianconi in 2021. They derive message passing type operators which pass information from edges to nodes and vice versa. From there they derive graph dynamics and a layerwise update rule which discretizes the time domain and adds in learnable weight matrices.

Network architecture consists of linear layer, then use DB layers with skip connections followed by a final MLP. They demonstrate good results compared to a couple of baselines in limited experiments.

**Strengths:**

Going beyond the vanilla MPNN approach and avoiding related oversmoothing problems is important and the authors provide an interesting setting, optimizing power grids, for needing to go deeper than standard DNNs. The approach is both well motivated from physics and novel in the context of GNNs.

The paper is generally well-written and well-explained.

**Weaknesses:**

The alternating procedure of DB and linear layers could be more clearly explained.  What does a linear layer mean in this context and why are they needed

Neither of the datasets considered inlcude non-trivial edge features. Perhaps you could have flux along an edge in a powergrid or something?

Not clear how the baseline methods where chosen? Are these also methods meant to address oversmoothing and long range dependencies? Are they at or near s-o-t-a?

x' and e'_{i,j} are not defined in equations 6 and 7. I am assuming they mean the output of delta_{DB}(x, e), but this should be made more clear

The setup assumes a symmetry on the edge features e_{j,i} = e_{i,j}

The leap from 8 to 9 should be made more clear. This appears to solving \partial_t - \partial_{DB} \pm \beta =0, or something but this should be made explicit.

Table 2 is interesting, that one DBGNN layer = 3 GCN layers, but it would be more convincing if you also showed results with DBGNN outpeforming GCN in addition to other baselines

**Questions:**

Is there any connection by the DB equation and the wave equation?

Is there any way to use the DB operator to define Riesz transforms ($\partial_{x_i}\Delta^{-1/2}$ in the Euclidean setting)?

Where does the network include non-linearitiees? Why do these not induce loss of energy?

What is the computational complexity of DBGNN vs other methods?

Paper could be improved via more thorough experimentation and also analysis of / references too the theoretical properties of the DB operator.

---

> ### Author Response · Authors · 2023-11-18
> **Reply to Reviewer 3z7c - Part 1**
>
> ```
> The alternating procedure of DB and linear layers could be more clearly explained. What does a linear layer mean in this context and why are they needed
> ```
>
> Linear layer simply refers to a learnable linear weight matrix of an appropriate dimension. We use them here for skip connections, that is, after running the DB dynamics for some time we allow the DBGNN to mix the features into the hidden dimensions again. This is expected to improve the ability of the system to go deep. We will expand on this feature further in the manuscript.
>
> Does this address the reviewers concerns, or were they also asking for clarification of other aspects of the DBGNN?
>
> ```
> Neither of the datasets considered inlcude non-trivial edge features. Perhaps you could have flux along an edge in a powergrid or something?
> ```
>
> We experimented with using the power flow on the edges as additional input features, but it did not improve the performance. See Reply to all Reviewers - Further Datasets.
>
> ```
> Not clear how the baseline methods where chosen? Are these also methods meant to address oversmoothing and long range dependencies? Are they at or near s-o-t-a?
> ```
>
> We have chosen the best performing models on the datasets as baselines, so we consider them state of the art. ARMANet and TAGNet, which perform best for the power grid task, share some features with architectures designed specifically to avoid oversmoothing, and are iterated quite deeply to achieve their performance. The DBGNN layer that beats them here is comparatively simpler, at least from a dynamics point of view.
>
> ```
> x' and e'{i,j} are not defined in equations 6 and 7. I am assuming they mean the output of delta{DB}(x, e), but this should be made more clear
> ```
>
> Thank you for pointing this out, you are right, we will clarify this point.
>
> ```
> Is there any connection by the DB equation and the wave equation?
> ```
>
> This is an excellent question. The Dirac equation is a wave equation, however, it includes an overall factor $i$ in front of the right hand side. As we omit this factor, the equation we use is not directly a wave equation. Instead, we have exponentially growing and shrinking modes. We will clarify this point in the manuscript.
>
> ```
> The leap from 8 to 9 should be made more clear. This appears to solving \partial_t - \partial_{DB} \pm \beta =0, or something but this should be made explicit.
> ```
>
> Equations 8 and 9 together are the Bianconi's topological Dirac equation written out explicitly in coordinates. We are not solving anything, edges and nodes evolve at the same time. We will clarify this point.
>
> ```
> Table 2 is interesting, that one DBGNN layer = 3 GCN layers, but it would be more convincing if you also showed results with DBGNN outpeforming GCN in addition to other baselines
> ```
>
> The 3 layer GCN is the state of the art for this task, see Reply to all Reviewers - Baselines for binding affinity. We are investigating further Datasets to strengthen the results of the paper, see Reply to all Reviewers - Further Datasets.
>
> ```
> Is there any way to use the DB operator to define Riesz transforms (in the Euclidean setting)?
> ```
>
> This is an interesting question, we do not know. Generally speaking, Dirac operators tend to require internal spaces and do not directly operate on real valued functions. We expect that any connection would be subtle.
>
> ```
> Where does the network include non-linearitiees? Why do these not induce loss of energy?
> ```
>
> We include a ReLu after every step of the dynamics. It is unclear to us why a non-linearity should be expected to induce a loss of Dirichlet energy. We will update the section on Dirichlet energy to better represent typical conditions in the layer, in which case we do see a gradual decline, see Reply to all Reviewers - Dirichlet Energy.
>
> ```
> What is the computational complexity of DBGNN vs other methods?
> ```
>
> The time complexity of the forward pass is the same as that of Message Passing Neural Networks. The main difference to these, is the fact that we have the edges as full updating states of the layer iteration. This does not increase the computational complexity. We will include a small note in the manuscript to this effect.

---

> > ### Author Response · Authors · 2023-11-18
> > **Reply to Reviewer 3z7c - Part 2**
> >
> > ```
> > Paper could be improved via more thorough experimentation and also analysis of / references too the theoretical properties of the DB operator.
> > ```
> >
> > We will include a few more pointers to the literature on the boundary operator in the context of machine learning. The major point of our work is not the introduction of the boundary operator, but of Bianconi's Dirac equation, with dynamical edges and mass terms. We will clarify this point. We believe that Bianconi already provided a fairly thorough analysis of the dynamics of the Dirac equation on networks, but the addition of higher dimensional feature spaces will change the picture substantially.
> >
> > We agree strongly that more experiments are needed to gain a better understanding of those layers properties, as well as the underlying dynamics, and will include more in the revised manuscript. See Reply to all Reviewers - Evidence for long range Capabilites and Further Datasets.

---

> > > ### Author Response · Authors · 2023-11-22
> > > **Second reply to Reviewer 3z7c**
> > >
> > > We want to thank Reviewer 3z7c for carefully reading our replies and increasing the score. Are there are any further questions?

---

> > ### Comment · Reviewer_3z7c · 2023-11-22
> >
> > I thank the authors for answering my questions

---

### Official Review · Reviewer_NtG7 · 2023-10-28

**Soundness:** 2 fair
**Presentation:** 2 fair
**Contribution:** 3 good
**Rating:** 5
**Confidence:** 3

**Summary:**

Summary:
The paper presents a new graph neural network layer called the Dirac-Bianconi Graph Neural Network (DBGNN). The layer is inspired by the topological Dirac equation on graphs proposed by Bianconi (2021). In DBGNN, the edges and nodes are treated on an equal footing, with features attached to both and the Dirac operator mixes edge and node features. Additionally, DBGNN avoids over-smoothing problem that affects traditional GNNs like GCNs.

The DBGNN layer implements a discretized, generalized version of the Dirac-Bianconi equation. Multiple DBGNN layers with shared weights are stacked to enable propagation across longer distances in the graph.

The DBGNN is evaluated on two tasks: i) Predicting power grid stability. (ii) Predicting protein-ligand binding affinity
In (i) it outperforms previous approaches, especially for out-of-distribution generalization. While in (ii), DBGNN achieves par with deeper GCNs.

In summary, this work presents a new GNN layer that can be helpful for long range tasks in graphs with some demonstration shown in the paper.

**Strengths:**

Strengths:
(i) well motivated combination of ideas from physics and GNNs. The Dirac equation is a natural model for directional propagation on graphs.
(ii) good empirical results on two relevant graph tasks. Outperforms prior models on power grids.
(iii) experiments analyzing model dynamics and over-smoothing.

**Weaknesses:**

Weaknesses and questions:
i) Limited ablation studies. It can be made further clear for how much performance gain comes from Dirac structure vs other enhancements.
ii) the paper does not exploit edge features on binding affinity task. It may be unsure if Dirac structure helps here.
iii) although one of the experimental task is related to long range dependency, it is not sure for the second task. A robust study on some long range dataset benchmark would be required to verify holistically the claims of the model on long range tasks as is mentioned in multiple instances in the paper. The paper would be strengthened by a more in-depth analysis of what long-range effects the model is capturing in this domain.
iv) For binding affinity task, did you try using edge features? Does the Dirac structure help in that case?

**Questions:**

above

---

> ### Author Response · Authors · 2023-11-18
> **Reply to Reviewer NtG7**
>
> ```
> i) Limited ablation studies. It can be made further clear for how much performance gain comes from Dirac structure vs other enhancements.
> ```
>
> It is unclear to us what the reviewer views as important other enhancements. DBGNN is a relatively simple layer. It is the iterated application of equations (11) and (12), which are the direct analogies of the Dirac Bianconi equation, followed by a skip connection. We have added a direct comparison to a simple MPNN to clarify this, and we now show that the equivalent MPNN does not have comparable dynamics or long range feature propagation (Reply to all Reviewers - Evidence for long range Capabilities).
>
> A more detailed study on how performance varies with various hyperparameters would, of course, be highly interesting. We also believe the layer to be highly compatible with the vast array of architectural enhancements that have been explored for MPNNs and GCNs. We think such broad questions are beyond the scope of this initial paper, though, and we decided to focus our experimental resources elsewhere.
>
> ```
> ii) the paper does not exploit edge features on binding affinity task. It may be unsure if Dirac structure helps here.
> ```
>
> It is important to note that in the absence of edge input features, we still have a non-trivial hidden edge dynamics. Thus, the way the nodal information propagates is shaped by the Dirac structure in this case, too. We will clarify this point in the paper.
>
> Thus we consider the competitive performance on this task a strength of the layer. It shows that the underlying dynamics is interesting. We assume that useful edge features can only help to get higher performances. We are also exploring further datasets, see Reply to all Reviewers.
>
> ```
> iii) although one of the experimental task is related to long range dependency, it is not sure for the second task. A robust study on some long range dataset benchmark would be required to verify holistically the claims of the model on long range tasks as is mentioned in multiple instances in the paper. The paper would be strengthened by a more in-depth analysis of what long-range effects the model is capturing in this domain.
> ```
>
> We agree that a much broader study would be needed. We have extended the analysis beyond the Dirichlet energy by including example trajectories of a randomly initialized layer. See Reply to all Reviewers - Evidence for long-range capabilities.
>
> We also would like to point out that the power grid task is known to require long-range capabilities (Ringsquandl et al. (doi: 10.1145/3459637.3482464)), and that the state-of-the-art models (ARMANet and TAGNet) are quite deep. Together with the evidence from randomly initialized layers, we consider the conclusion reasonable that DBGNNs are good for long-range dependencies.
>
> At the same time, we agree that a study of DBGNN on a long-range dataset benchmark would strengthen the paper considerably, and we are working towards that goal. See also Reply to all Reviewers - Further Datasets.
>
> ```
> iv) For binding affinity task, did you try using edge features? Does the Dirac structure help in that case?
> ```
>
> See Reply to all reviewers (Baselines for binding affinity).

---

### Official Review · Reviewer_VKy2 · 2023-10-30

**Soundness:** 3 good
**Presentation:** 2 fair
**Contribution:** 2 fair
**Rating:** 3
**Confidence:** 4

**Summary:**

The paper introduces the Dirac-Bianconi Graph Neural Network (DBGNN) based on Bianconi's topological Dirac equation for graph-based network dynamics. DBGNN preserves long-range information propagation and treats edges and nodes equally. This approach is beneficial when edges convey physical properties. The paper demonstrates competitive performance in molecular property prediction and superior performance in predicting the dynamic stability of power grids. In the case of power grids, DBGNN exhibits robust out-of-distribution generalization, indicating learned structural relations.

**Strengths:**

The development of the Dirac-Bianconi Graph Neural Network (DBGNN) based on a modified generalized Dirac-Bianconi equation represents an innovative approach. The incorporation of learnable weights, nonlinearity, and multiple time steps for evolving features offers a unique solution to address long-range dependencies in graph neural networks. The integration of fundamental quantum equations into GNN research is relatively rare, demonstrating a level of innovation. The paper upholds a high standard of quality in its methodology, showcasing the effectiveness of the DBGNN architecture in challenging tasks, including power grid analysis and molecular property prediction. The model's performance underscores its robustness and practicality. The competitive performance and out-of-distribution generalization observed in power grid tasks, along with enhanced molecular property prediction, demonstrate the practical relevance of this work.

**Weaknesses:**

The theoretical part of this paper lacks an analysis of the time complexity of the Dirac-Bianconi Equation. The paper does not provide a detailed theoretical explanation of the integration of the Dirac-Bianconi equation with Graph Neural Networks (GNNs). It focuses on the properties of the Dirac-Bianconi equation itself, dedicating substantial space to this, but does not elaborate on the theoretical foundation for its fusion with GNNs.
The paper applies the DBGNN model to two vastly different tasks, power grid analysis and molecular property prediction, without tailoring the model for the specifics of each task. This lack of task-specific optimization could limit the model's performance and applicability. The experimental setup in section 4.1.2 appears to have some shortcomings in terms of training parameters and code implementation. This raises concerns about the reproducibility and robustness of the experiments, which are essential in scientific research.
The paper's comparison with other models, particularly using only GCN with three layers, is not comprehensive and may not represent a fair benchmark. Comparing models with varying numbers of layers and different architectural complexities would provide a more meaningful evaluation. Lack of clear improvement in accuracy with the use of the Dirac-Bianconi equation within GNNs could be seen as a weakness in the paper's claims regarding its advantages.

**Questions:**

Can a more detailed theoretical explanation be provided regarding why the Dirac-Bianconi equation was chosen for integration with GNNs and how they are combined to enhance graph representation capabilities?
Given the vastly different nature of power grid analysis and molecular property prediction, is there a plan to introduce task-specific optimizations within the model to leverage the unique characteristics of each task?
Could more detailed experimental settings and parameter choices be provided to ensure experiment reproducibility while also considering the robustness of the model?

---

> ### Author Response · Authors · 2023-11-18
> **Reply to Reviewer VKy2**
>
> ```
> [The paper] does not elaborate on the theoretical foundation for its fusion with GNNs.
> ```
>
> Many GCN methods are equivalent to dynamical systems on networks. This has been elaborated in the literature we cite in the introduction. Given this close equivalence it is unclear to us what theoretical foundation for the fusion the Reviewer had in mind. We are happy to elaborate if the question is clarified.
>
> ```
> The theoretical part of this paper lacks an analysis of the time complexity of the Dirac-Bianconi Equation.
> ```
>
> The time complexity of the forward pass is the same as that of Message Passing Neural Networks. The main difference to these is the fact that we have the edges as full updating states of the layer iteration. This does not increase the computational complexity. We have included a small note in the manuscript to this effect.
>
> ```
> The paper applies the DBGNN model to two vastly different tasks, power grid analysis and molecular property prediction, without tailoring the model for the specifics of each task. This lack of task-specific optimization could limit the model's performance and applicability.
> ```
>
> We agree that the performance of DBGNN could be further improved, but it already shows competitive and superior performance in comparison to the baselines. We consider it as a strength that the layer as presented is relatively simple but already exhibits interesting performance. Adding optimizations should therefore be possible, but we consider it beyond the scope of this paper, which focuses on the properties and performance of the core layer.
>
> ```
> The experimental setup in section 4.1.2 appears to have some shortcomings in terms of training parameters and code implementation.
>
> Could more detailed experimental settings and parameter choices be provided to ensure experiment reproducibility while also considering the robustness of the model?
> ```
>
> Details of the hyperparameters and used software are provided in the appendix. Furthermore, we will provide all code to make the results fully reproducible. We would be glad to add more details if Reviewer VKy2 elaborates on the type of missing information.
>
> We also tried different hyperparameters and show some examplary results of different configurations below:
>
>
> | learning rate | dropout_n | dropout_e | hidden dimension n| hidden dimension e | tr20ev20  |
> | --- | ---| ---| --- | ---- | ---|
> | 3.6e-4 | 6.1e-2 | 0| 500| 10 |84.79|
> | 1.3e-3 | 5.5e-2 | 0| 500| 10 |85.96|
> | 1.3e-3 | 6e-2 | 0| 500| 10 |85.73|
> | 2.2e-3 | 2e-2 | 0| 300| 10 |84.13|
> | 4.6e-4 | 5.5e-2 | 0| 250| 250 |84.35|
> | 2e-4 | 1.3e-2 | 1.3e-2| 300| 100 |80.87|

---

### Official Review · Reviewer_Bj5J · 2023-10-31

**Soundness:** 2 fair
**Presentation:** 2 fair
**Contribution:** 2 fair
**Rating:** 5
**Confidence:** 3

**Summary:**

This paper proposes a new Graph Neural Network, the Dirac-Bianconi Graph Neural Network, derived from an Euler discretization of the generalized Dirac-Bianci equation on a network. While the graph Laplacian operator-based GNNs cause over-smoothing, the proposed method is designed to capture long-range interactions between nodes. This paper confirms that the proposed method does not cause over-smoothing evaluated using Dirichlet Energy. Also, this paper applies the proposed method to estimating power grid stability and predicting binding affinity and compares its prediction accuracy with existing methods.

**Strengths:**

* The proposed method improves the accuracy of power grid stability prediction compared to existing methods, especially for the out-of-distribution problem setting.
* Numerical results show that the over-smoothing evaluated using Dirichlet Energy can be alleviated for the model trained on real data.
* The paper is well-written, and the derivation of the proposed model is carefully described, making it accessible to readers unfamiliar with the Dirac-Bianconi operator.

**Weaknesses:**

* If I understand correctly, the Dirac-Bianci operator is called the boundary and co-boundary operators in the simplicial complex theory on graphs. Existing studies propose GNNs that use or extend them (e.g., [1--4]. Also, [5] does not use (co-)boundary operators but extends GNNs on simplicial complexes.) Therefore, it is debatable whether the proposed method has (theoretical and experimental) novelty and significance in terms of using the Dirac-Bianci operator.
* This paper claimed that one of the advantages of the proposed method is that it solves over-smoothing. However, since there exist models that tackle over-smoothing, such as GCNII [6] and DRew [7], I have a question about whether the proposed method is superior to them.
* In the task of binding affinity prediction, this paper argues that the proposed method with ten steps is comparable with 3-layer GCN. However, the deeper the model is the greater the computational complexity and memory usage. Since existing models overcome the over-smoothing, as I mentioned above, more is needed for a deep model to be comparable in performance to a shallow GNN.
* One of the advantages of the proposed method is that both node and edge features can be used equally. However, in the numerical experiments on real data, only node features are available to GNNs (in the binding affinity prediction task, edge features are used for graph construction but not as features). Therefore, it is unclear whether the proposed method can take advantage of this feature in real data.

- [1] https://openreview.net/forum?id=vbPsD-BhOZ
- [2] https://proceedings.mlr.press/v139/bodnar21a
- [3] https://openreview.net/forum?id=ScfRNWkpec
- [4] https://proceedings.mlr.press/v139/roddenberry21a.html
- [5] https://openreview.net/forum?id=nPCt39DVIfk
- [6] https://proceedings.mlr.press/v119/chen20v.html
- [7] https://openreview.net/forum?id=WEgjbJ6IDN

**Questions:**

* How were the hyperparameters chosen? In particular, the numerical experiments use networks with 48 and 10 steps in total, respectively. However, it is yet to be known whether these hyperparameters are optimal. That is, a model with fewer steps might be sufficient. I suggest conducting ablation studies to see the sensitivity of performances to the number of steps.
* In Figure 5(a), the Dirichlet energy of untrained DBGNN is almost constant regardless of the number of steps. Is this expected from the theory?

Minor Comments
- P4, Section 2: $\partial_{db}$ -> $\partial_{DB}$
- Section A.3: pyTorch -> PyTorch

**Details Of Ethics Concerns:**

N.A.

---

> ### Author Response · Authors · 2023-11-18
> **Reply to Reviewer Bj5J - Part 1**
>
> ```
> If I understand correctly, the Dirac-Bianci operator is called the boundary and co-boundary operators in the simplicial complex theory on graphs. Existing studies propose GNNs that use or extend them (e.g., [1--4]. Also, [5] does not use (co-)boundary operators but extends GNNs on simplicial complexes.) Therefore, it is debatable whether the proposed method has (theoretical and experimental) novelty and significance in terms of using the Dirac-Bianci operator.
> ```
>
> We thank the reviewer for highlighting the connection to this part of the literature. The comment was extremely helpful. We agree that the boundary operator in itself is not new, and we cite Loyd et al. as an early work on using the boundary operator in the context of topological data analysis on simplicial complexes.
>
> The key point of Bianconi's Dirac equation is the combination of the boundary operator, and having the edge degree of freedoms as full participants in the dynamics with an appropriate mass term. This causes a spectral gap, and interesting propagation on the network. The work \[1\] on sheaf diffusion for example, while considering a very similar topological set up as we do, does not treat the edge space as possessing its own dynamics, and thus cannot induce analogous dynamics.
>
> The topological set up is of the general form considered in \[2\], which we will note in the manuscript. However, in \[2\] the focus is on simplicial complexes, and in their applications to graphs, the set up we use is not considered (we have boundary and coboundary at the same time, but no up/down/Hodge Laplacians. The set ups of \[2\] did not feature the combination of a boundary/coboundary operator). In a further literature review based on the input of the Referee we identified ( arxiv.org/abs/2012.06010 ) as one example where a similar set up appears, and we will cite this as a predecessor.
>
> Overall we believe our approach is closer to, e.g. the use of Kuramoto oscillators and other physically inspired dynamical equations that are not diffusive, rather than approaches inspired from algebraic topology. The close relationship to these works is of course highly relevant and of great theoretical interest. We will rework the theory section to make this clear.
>
> ```
> This paper claimed that one of the advantages of the proposed method is that it solves over-smoothing. However, since there exist models that tackle over-smoothing, such as GCNII [6] and DRew [7], I have a question about whether the proposed method is superior to them.
> ```
>
> The motivation to explore the use of the DB equation was twofold, we wanted to be able to deal with edge features and to use a non-diffusive dynamical equation that would enable the deep predictions needed for power grid analysis. In our view, it is highly interesting that the DB equation possesses these qualities without having to be specifically engineered for it.
>
> We agree that it would be interesting to understand in more detail how it compares to other approaches that were designed to handle oversmoothing. In the power grid context, deep GNNs such as ARMANets and TAGNets are known to have the best performance. These share some characteristics with GCNII. DBGNN is a relatively simple layer, that does not make use of these strategies, yet has competitive performance in this dataset at least.
>
> It would be highly interesting to see whether these strategies, as well as those used in DRew, can be used to further improve the performance of DBGNN. We will add this as a research question to the outlook, but consider it out of scope for this paper.
>
> ```
> this paper argues that the proposed method with ten steps is comparable with 3-layer GCN [...] to be comparable in performance to a shallow GNN.
>
> One of the advantages of the proposed method is that both node and edge features can be used equally. However, in the numerical experiments on real data, only node features are available to GNNs (in the binding affinity prediction task, edge features are used for graph construction but not as features). Therefore, it is unclear whether the proposed method can take advantage of this feature in real data.
> ```
>
> We have expanded on the baseline comparison, and agree that more diverse datasets would be desirable, see the reply to all reviewers.
>
> ```
> How were the hyperparameters chosen? In particular, the numerical experiments use networks with 48 and 10 steps in total, respectively. However, it is yet to be known whether these hyperparameters are optimal. That is, a model with fewer steps might be sufficient. I suggest conducting ablation studies to see the sensitivity of performances to the number of steps.
> ```
>
> We conducted unstructured hyperparameter studies testing different number of steps. We believe that there is more potential when conducting more hyperparameter studies, but even with our small studies, we could already outperform current baselines.

---

> > ### Author Response · Authors · 2023-11-18
> > **Reply to Reviewer Bj5J - Part 2**
> >
> > We also thank the Reviewer Bj5J for the minor comments and fixed it in the updated version.
> >
> > ```
> > In Figure 5(a), the Dirichlet energy of untrained DBGNN is almost constant regardless of the number of steps. Is this expected from the theory?
> > ```
> >
> > See Reply to all reviewers (Dirichlet Energy).

---

> > > ### Comment · Reviewer_Bj5J · 2023-11-22
> > > **Response to Authors' Comments**
> > >
> > > I thank the authors for answering my review comments. Here, I respond to the authors' comments one by one.
> > >
> > > ### Comparison with GNNs inspired by algebraic topology theory on graphs
> > >
> > > > The key point of Bianconi's Dirac equation is the combination of the boundary operator, and having the edge degree of freedoms as full participants in the dynamics with an appropriate mass term. This causes a spectral gap, and interesting propagation on the network. The work [1] on sheaf diffusion for example, while considering a very similar topological set up as we do, does not treat the edge space as possessing its own dynamics, and thus cannot induce analogous dynamics.
> > >
> > > I understand that the proposed method differs from [1] in that it can incorporate edge features.
> > >
> > > ------------------------
> > >
> > > > The topological set up is of the general form considered in [2], which we will note in the manuscript. However, in [2] the focus is on simplicial complexes, and in their applications to graphs, the set up we use is not considered (we have boundary and coboundary at the same time, but no up/down/Hodge Laplacians. The set ups of [2] did not feature the combination of a boundary/coboundary operator).
> > >
> > > I understand that Message Passing Simplicial Network (MPSN) proposed in [2] is a general form of message passings between simplicial objects. Also, Simplicial Isomorphism Network (SIN), a specific realization of MPSN, does not use boundary and co-boundary operators.
> > >
> > > ### Comparison with GNNs for over-smoothing
> > >
> > > > We agree that it would be interesting to understand in more detail how it compares to other approaches that were designed to handle oversmoothing. In the power grid context, deep GNNs such as ARMANets and TAGNets are known to have the best performance. These share some characteristics with GCNII. DBGNN is a relatively simple layer, that does not make use of these strategies, yet has competitive performance in this dataset at least.
> > >
> > > Certainly, ARMANet has initial residuals as in GCNII, as shown in Eq. (14) of [A]. However, GCNII has other characteristics, such as the identity mapping and hyperparameter tuning of $\beta_l$, which also help to mitigate the over-smoothing problem. Regarding TAGNet, looking at [B], especially Eq. (5), it is not easy to think that TAGNet has GCNII's characteristics in common. In addition, although the power grid task in Section 4.1 is a problem of estimating the dynamic system since it is a node prediction task on a graph, any GNN that provides embedding for each node can be applied. Therefore, I think the fact that the task is a power grid estimation may not imply that we do not have to compare the proposed model with the SOTA models for over-smoothing to claim that it solves the over-smoothing problem.
> > >
> > > [A] https://arxiv.org/abs/1901.01343
> > > [B] https://arxiv.org/abs/1710.10370
> > >
> > > ### Baselines for binding affinity
> > >
> > > > The surprising fact that 3 layers might be sufficient is probably an indicator that long-range dependencies may not be relevant in that case. Furthermore, the dataset does not contain edge features. Those factors probably limit the chances of DBGNN to show advantages over GCN.
> > >
> > > I thank the authors for clarifying the motivation for choosing 3-layer MLP as a baseline. However, if my understanding is correct, the model proposed as an answer to the research question of whether long-range interactions can be adequately captured. This explanation raises whether predicting binding affinity is an appropriate dataset for this research question. I want to confirm what the authors want to claim from this task.
> > >
> > > ### Constant Dirichlet energy
> > >
> > > > In the previously submitted version, the weights were initialized too small. With larger weights, the growing and shrinking modes start to mix, and we obtain a more complex behavior of the Dirichlet Energy, as might be expected. We will swap the figure for one with weights drawn from a wider distribution.
> > >
> > > OK. I look forward to the additional experiment results using the wider weight distributions.

---

> > > > ### Author Response · Authors · 2023-11-22
> > > > **Second reply to Reviewer Bj5J**
> > > >
> > > > We thank the Reviewer for the detailed response and want to clarify details.
> > > >
> > > > ` I understand that the proposed method differs from [1] in that it can incorporate edge features.`
> > > >
> > > > We agree, just to be certain, DBGNN does not only allow dealing with edge features, but it also dynamically treats the propagation of edge features, whereas other GNNs only propagate node features.
> > > >
> > > > `  Regarding TAGNet, looking at [B], especially Eq. (5), it is not easy to think that TAGNet has GCNII's characteristics in common.  [...] Therefore, I think the fact that the task is a power grid estimation may not imply that we do not have to compare the proposed model with the SOTA models for over-smoothing to claim that it solves the over-smoothing problem.`
> > > >
> > > > TAGConv and iteration of GCNII layers as provided by PyTorch \[1, 2\] both generate polynomials in $A$. This is not the case for vanilla GCNs which only contain the highest power. As TAGConv has general coefficients its weight space contains GCNII. For example, collecting weights, \\alpha and \\beta into matrices W and iterating twice:
> > > >
> > > > H_1 =A H_0 W_10 + H_0 W_1
> > > > H_2 = A H_1 W_21 + H_0 W_2
> > > > = A (A H_0 W_10 + H_0 W_1) W_21 + H_0 W_2
> > > > = A^2 H_0 W_10 W_21 + A H_0 W_1 W_21 + H_0 W_2
> > > >
> > > > The fact that GCNII convolutions are contained in the weight space of TAGConv does not imply that when adding nonlinearities, or with respect to gradient descent, the networks are equivalent. One would certainly expect very different gradients. But this is beyond the scope of what we can discuss here. As TAGNet was successful for the power grid dataset, it must be capable of some long-range behavior.
> > > >
> > > > Unfortunately, the authors of the PowerGrid model did not provide a study of GCNII, and we did not have resources now to do so. However, we now include a study of a long range benchmark dataset that includes GCNII.
> > > >
> > > > The claim regarding the over-smoothing problem is also based on the theoretical analysis of the Dirichlet energy, and the spectral properties of the DB equation.
> > > >
> > > > ` I want to confirm what the authors want to claim from this task.` After showing great performances on power grid tasks, we wanted to apply DBGNN in a very different setting, and we chose the binding affinity prediction. We wanted to make sure, that DBGNN can not only be applied in the context of power grids. We fully agree with the referee that this task does not contribute to the evidence for long range capabilities. For this other datasets would have been more appropriate, hence we also provide performances on the long range molecular task.
> > > >
> > > > \[1\] <https://pytorch-geometric.readthedocs.io/en/latest/generated/torch_geometric.nn.conv.GCN2Conv.html#torch_geometric.nn.conv.GCN2Conv>
> > > >
> > > > \[2\] <https://pytorch-geometric.readthedocs.io/en/latest/generated/torch_geometric.nn.conv.TAGConv.html#torch_geometric.nn.conv.TAGConv>

---

> > > > > ### Comment · Reviewer_Bj5J · 2023-11-23
> > > > >
> > > > > I thank the authors for the detailed responses. I understand what the authors intended to mean. I will consider the evaluation based on the authors' comments.

---

### Official Review · Reviewer_YvSy · 2023-11-03

**Soundness:** 3 good
**Presentation:** 2 fair
**Contribution:** 2 fair
**Rating:** 3
**Confidence:** 4

**Summary:**

Deviating from the traditional Laplacian based GNNs, on this paper the authors propose Dirac-Bianconi Graph Neural Network (DBGNN) which are based on the topological Dirac equation on the graph. The major advantage of the proposed method is that it does not lead to over-smoothing of the node features when a large number of layers are stacked.


The paper is difficult to read at times. Especially for readers not having a background in topological data analysis. The paper is well motivated but lacks insights when using Dirac-Bianconi operators. Although Figure 5 demonstrate that the proposed method enabling heterogeneous representations even after 500 layers, giving more insights will help improving the paper.

**Strengths:**

1. The work is well motivated and the main advantage seems to avoid over-smoothing and also use input edge features in learning representations.

2. Figure 5 is a good demonstration of avoiding over-smoothing.

3. Slight performance improvement on power grid data.

**Weaknesses:**

1. In abstract, you mention "we expect DBGNN to be useful in contexts where edges encode more than mere logical connectivity, but have physical properties as well". Also the introduction motivates for the same. However, the datasets used in the evaluation do not consider any useful edge information. Including datasets with useful input edge features will enrich the paper.

2. In Table 2, it is shown that only 1 layer of DBGNN suffices. This is deviating from the story that stacking more layers helps in complex (hard) datasets.

I am willing to raise my score significantly if the story of the paper is in line with the findings and more insights with toy examples are presented.

**Questions:**

1. Why do we need at least 13 layer GNNs for power grids? Is it true for all type of GNNs? Some powerful GNNs might need fewer than 13 layers.

2. Please clarify using the notation $e_{ij}$ = $ - e_{ji}$.

3. What is the performance of the proposed method on node classification tasks on citation networks such as Cora, Pubmed, etc?

4. The Dirichlet energy is constant over 500 steps in Figure 5? Why? Is it just for this experiment? It does not seem to change a t all..

---

> ### Author Response · Authors · 2023-11-18
> **Reply to Reviewer YvSy**
>
> ```
> However, the datasets used in the evaluation do not consider any useful edge information.
> ```
>
> We agree. See also Reply to All Reviewers - Further Datasets. We will clarify that even in the absence of edge features, the DB equations lead to a very different behavior than standard MPNNs, see Reply to All Reviewers - Evidence for long-range capabilities of DBGNN.
>
> ```
>  it is shown that only 1 layer of DBGNN suffices
> ```
>
> This is probably a property of the dataset. If 1 layer with 12 steps already considers sufficiently large subgraphs, more layer will not be necessary.
>
> ```
> Why do we need at least 13 layer GNNs for power grids? Is it true for all type of GNNs? Some powerful GNNs might need fewer than 13 layers.
> ```
>
> Ringsquandl et al. (doi: 10.1145/3459637.3482464) state that the unique structure of power grids leads to the need at least 13 layers for GNNs. Even that might not be sufficient for all tasks: the behavior of power grids shows profound non-locality. A failure in one part of the grid can cause a subsequent failure at a distance comparable to the system size (see e.g. <https://www.science.org/doi/10.1126/science.aan3184>). For example, large system splits and blackouts in Southern Europe have been caused by line failures in Northern Germany. The specific task we investigate is not known to be quite so non-local, but features the same physics.
>
> ```
> Please clarify using the notation e_ij = - e_ji .
> ```
>
> See reply to all Reviewers - Edge notation.
>
> ```
> What is the performance of the proposed method on node classification tasks on citation networks such as Cora, Pubmed, etc?
> ```
>
> We do not expect that DBGNN has an advantage over existing methods on citation networks. Large topological patterns are not known to be relevant in this case, while close neighborhoods play a more significant role. See also Reply to All Reviewers - Further Datasets.
>
> ```
> The Dirichlet energy is constant over 500 steps in Figure 5? Why? Is it just for this experiment? It does not seem to change a t all..
> ```
>
> See Reply to all reviewers - Constant Dirichlet energy.
>
> ```
> I am willing to raise my score significantly if the story of the paper is in line with the findings and more insights with toy examples are presented.
> ```
>
> We are more than happy to provide further support for the story line of the paper. We will add a visualization of the long range response of the layer to a localized signal. See Reply to all Reviewers.  If the reviewer has further toy examples that they would find informative, we would be more than happy to implement them.
>
> As the revision of the manuscript will still take some days we have already included the new visualization in a supplementary note.

---

### Author Response · Authors · 2023-11-18
**Reply to all Reviewers - Part1**

We want to thank all reviewers for their time and effort they invested to review our paper and appreciate that they highlighted the strengths: "well motivated", "good demonstration of avoiding over-smoothing" (YvSy), "paper is well-written", "making it accessible to readers" (Bj5J), "innovative approach", "paper upholds a high standard of quality" (VKy2), "well motivated combination of ideas from physics and GNNs", "good empirical results" (NtG7), "The paper is generally well-written and well-explained" (3z7c).

The reviewers have highlighted several areas that merit improvement and open research questions. We thank the reviewers for their constructive feedback. First, we address issues raised by multiple reviewers, before providing individual replies.

As the manuscript preparation will still take a few days, we have already included the updated and new figures with these replies. They can be found in the supplementary materials in the pdf with the name: ReplyToReviewersNewImages.pdf.

## Constant Dirichlet energy

Several researches asked why the Dirichlet stays constant in case of DBGNN with many steps. For weights close to the identity $W = I$, the long-term trajectory of the equations is dominated by the dominant eigenvectors of the linear Dirac Bianconi equation. For the Laplacian this is a homogeneous mode and differences smooth out. The DB equation, as analyzed by Bianconi, instead has many growing and shrinking modes that are highly heterogeneous.

In the previously submitted version, the weights were initialized too small. With larger weights, the growing and shrinking modes start to mix, and we obtain a more complex behavior of the Dirichlet Energy, as might be expected. We will swap the figure for one with weights drawn from a wider distribution. This is more informative for what the layer actually does after training as well. The new figure can be found in the supplementary material and will be included in the updated manuscript.

## Evidence for long-range capabilities of DBGNN

Several reviewers asked for further investigations of the long-range capabilities of DBGNN. To do so, we randomly initialized a DBGNN and a Message Passing Network (MPNN) with the same weights. We then plotted the excitations of the hidden features vs steps, starting from an input that is zero everywhere except at node 1, where it is chosen at random.

We find that a randomly initialized DBGNN layer can travel the complete length of a path graph without diffusion, while an MPNN with the same weights diffuses into the graph. We have included the plots for this behavior in the supplementary information and will incorporate them into the manuscript.

## Further Datasets

We acknowledge the legitimacy of the request to add other datasets that incorporate more interesting edge features and low range dependencies. We have started investigating a long range benchmark task with edge features suggested by Reviewer NtG7, and hope to be able to report on this before the end of the discussion period, but will certainly include results of this investigation in the final version of the manuscript.

Initially, our motivations for exploring DBGNN were to incorporate edge features in an architecture that is capable of long range interactions. The edge features we wanted to use were the power flows in the power grid tasks. We found that DBGNN and other layers that incorporate edge information don't improve when including the power flow. We consider it highly likely that this is a property of the dataset. As this discussion has no direct relationship to DBGNN we believe it is not relevant to this paper. We consider the fact that we could improve on the best known architecture for the power grid task, even in the absence of edge features, a promising sign.

Furthermore, neither citation networks nor social graphs are expected to include long range interactions or strong edge features. We therefore decided to focus on other tasks in this work.


## Baselines for binding affinity

Multiple reviewers raised questions regarding the binding affinity prediction task and the reason to compare DBGNN to GCN. We decided to use 3 layer GCN, because it currently holds the best performance on this dataset. The surprising fact that 3 layers might be sufficient is probably an indicator that long-range dependencies may not be relevant in that case.  Furthermore, the dataset does not contain edge features. Those factors probably limit the chances of DBGNN to show advantages over GCN. The 3-layer GCN and DBGNN have a comparable number of parameters and training times. The 3-layer GCN model has approximately 1,693,525 trainable parameters, whereas the DBGNN model has around 1,775,745. Both models require similar training times, with each taking approximately 3 hours to complete 500 epochs on Nvidia A100 GPU with 80GB RAM.

---

> ### Author Response · Authors · 2023-11-18
> **Reply to all Reviewers - Part 2**
>
> ## Edge notation
>
> Reviewers YvSy and 3z7c raised questions regarding the edge notation:
>
> ```
> The setup assumes a symmetry on the edge features e_{j,i} = e_{i,j}
> ```
>
> ```
> Please clarify using the notation e_ij = - e_ji .
> ```
>
> The Dirac Bianconi equation assumes an undirected graph, and a fiducial orientation on the edges. The signs in the incidence matrix $B$ depend on this choice. The edge values for an edge are associated with the fiducial orientation $e\_{ij}$. The values associated to the opposite of the fiducial orientation are $-e\_{ji}$. These conventions result in the usual definition of the graph Laplacian of an undirected graph.
>
> In the present context it is clearly more appropriate to treat $e\_{ij}$ and $e\_{ji}$ separately, as is typically done in the context of MPNNs. We will adjust the theoretical presentation accordingly to remove the fiducial orientation.

---

> > ### Author Response · Authors · 2023-11-22
> > **Provide results on new long range dataset with edge features**
> >
> > As mentioned previously, we would like to report more results on a suggested dataset from the long-range benchmark. We tested DBGNN on the Peptides-struct dataset from (https://doi.org/10.48550/arXiv.2206.08164). The proposed DBGNN contains only 63911 parameters in comparison to the benchmark models which have roughly 500 000 parameters. DBGNN significantly outperforms the GCN, modified GCNs for long-range datasets such as GCNII, GatedGCN as well as GIN. Furthermore, we provide the transformer-based performances below. Transformer-based architectures achieve higher performances than GCN and DBGNN.  However, future work may incorporate attention-based methods in DBGNN, to enable even higher performances, but we consider this out of scope for this paper. The results of the DBGNN model are preliminary, because they are not fully converged yet. We expect the final results to be even better.
> >
> > | Model | test MAE |
> > |-------|----------|
> > | GCN | 0\.3496±0.0013 |
> > | GCNII | 0\.3471±0.0010 |
> > | GINE | 0\.3547±0.0045 |
> > | GatedGCN | 0\.3420±0.0013 |
> > | GatedGCN + RWSE | 0\.3357±0.0006 |
> > | DBGNN | 0\.3288 ± 0.0046 |
> >
> > Transformer-based models:
> >
> > | Model | test MAE |
> > |-------|----------|
> > | Transformer+LapPE | 0\.2529±0.0016 |
> > | SAN+LapPE | 0\.2683±0.0043 |
> > | SAN+RWSE | 0\.2545±0.0012 |
> >
> > In case there are still open questions, we are more than happy to address them. We will also upload an updated manuscript before the deadline.

---

> > > ### Author Response · Authors · 2023-11-22
> > > **Submission of revised paper**
> > >
> > > We would like to thank all reviewers for their very helpful and constructive points. We have substantially extended the paper, with a strong focus on the long-range capabilities. The most important updates are:
> > >
> > > - New Dataset: We present competitive performance to other GCNs on peptides-struct from the Long-Range Benchmark Dataset with a roughly 10x smaller model.
> > >
> > > - Discussion of wave nature of the equations and spectral properties of the linear dynamics: We discuss explicitly under which conditions the layer can be expected to show wave like and expanding/contracting dynamics.
> > >
> > > - Evidence of long-range traveling activation waves from toy models: We show that the wave nature of the equations enables coherent spreading of activation waves deep into the graph. The non-linearities on edges, and other features are seen to play no major role as an MPNN with the same weights and features does not exhibit this behavior.
> > >
> > > We again would like to thank the reviewers for enabling this substantial improvement through their valuable feedback.

---

### Meta-Review · Area_Chair_22U8 · 2023-12-05

**Metareview:**

This paper introduces "Dirac-Bianconi GNN", a graph neural net architecture designed to avoid in particular over-smoothing. The referees saw several issues: 1. the theoretical underpinnings are not sufficiently explored or explained. 2. the experimental setup and the choice of datasets do not convincingly demonstrate the claimed advantages of the DBGNN.  3. the paper lacks a comprehensive comparison with existing models that address similar issues.

**Justification For Why Not Higher Score:**

No compelling comparison with existing models that address over-smoothing.

**Justification For Why Not Lower Score:**

N/A

---

### Decision · Program_Chairs · 2024-01-16

Reject